# Deconvolution volumetric additive manufacturing

Antony Orth [1,4] ✉, Daniel Webber[1,4] ✉, Yujie Zhang[1], Kathleen L. Sampson [1], Hendrick W. de Haan[2], Thomas Lacelle [1], Rene Lam[1], Daphene Solis [1], Shyamaleeswari Dayanandan[1], Taylor Waddell[3], Tasha Lewis[3], Hayden K. Taylor [3], Jonathan Boisvert[1] & Chantal Paquet [1]

Volumetric additive manufacturing techniques are a promising pathway to ultra-rapid light-based 3D fabrication. Their widespread adoption, however, demands significant improvement in print fidelity. Currently, volumetric additive manufacturing prints suffer from systematic undercuring of fine features, making it impossible to print objects containing a wide range of feature sizes, precluding effective adoption in many applications. Here, we uncover the reason for this limitation: light dose spread in the resin due to chemical diffusion and optical blurring, which becomes significant for features $\lesssim 0.5$ mm. We develop a model that quantitatively predicts the variation of print time with feature size and demonstrate a deconvolution method to correct for this error. This enables prints previously beyond the capabilities of volumetric additive manufacturing, such as a complex gyroid structure with variable thickness and a fine-toothed gear. These results position volumetric additive manufacturing as a mature 3D printing method, all but eliminating the gap to industry-standard print fidelity.

In additive manufacturing (AM), objects are generally fabricated one voxel, line, or layer at a time. This paradigm has been upended recently by the introduction of volumetric additive manufacturing (VAM) techniques, where the entire volume is printed simultaneously[1,2]. The most widely used VAM technique leverages tomographic principles to project a 3D light dose inside a rotating vial containing a photo-sensitive resin. When and where the local light dose exceeds the polymerization threshold, the object cures, while the rest of the print volume remains liquid. In this manner, ~10–20 mm objects are typically printed on a timescale of ~10 s–1 min –an enormous speed improvement over traditional voxel/line/layer-based approaches. Moreover, because cured objects are created while suspended in a nearly density-matched surrounding of the uncured resin, support structures are not needed. However, this gain in print speed and design freedom comes with challenges. Unlike other vat photopolymerization methods such as raster scanning stereolithography (SLA) or digital light processing (DLP) printing, light exposure is not restricted to a single layer or voxel

in VAM. Light dose is applied everywhere in the volume, increasing the likelihood of overexposure. Moreover, in VAM the print volume is continuously illuminated for tens of seconds to over a minute before polymerization, compared to ~0.01–0.1 s/voxel in SLA and ~1–10 s/layer in DLP, leaving more time for diffusive effects to manifest.

Recently, we introduced an imaging technique called "optical scattering tomography" (OST) to observe and quantify the 3D poly-merization process in VAM in real-time[3]. In this previous paper, we remarked on systematic differences in print time (indicated by the onset of optical scattering) between large and small features. A brief literature review indicates that this is indeed a general phenomenon in VAM[2–7]: large features tend to polymerize first (or are overcured), followed by small features that are underexposed, if present at all. For example, the ears of the Stanford Bunny (small features) in[5] are shorter than the reference design; fine features of the Maitreya model in[4] are absent; the steering wheel dimple and rear tube of the Benchy model in[2] do not form, nor do the inner cogs fully form in the conventional

[1]National Research Council of Canada, Ottawa, ON, Canada. [2]Ontario Tech University, Oshawa, ON, Canada. [3]University of California Berkeley, Berkeley, CA, USA. [4]These authors contributed equally: Antony Orth, Daniel Webber. ✉e-mail: antony.orth@nrc-cnrc.gc.ca; daniel.webber@nrc-cnrc.gc.ca

VAM example in[6]. In a 3-beam VAM setup, this effect was reported for strut structures[7]. Previous dynamic observations of VAM polymerization also show this effect: the order of appearance of Benchy features in[3]; the polymerization of the large sphere before the small sphere in Supplementary Movie 1 in Loterie et al.[2], and polymerization of the object in Supplementary Movie 2 in Loterie et al.[2], starting from the thick middle region, ending at the thin top and bottom regions.

In this paper, we demonstrate this size-dependent polymerization time with a simple printing experiment involving disks of varying thickness. We then proceed to identify the cause of this feature-size dependent polymerization time as a combination of time-dependent dose diffusion and the projector's optical point spread function (PSF). Using OST, we directly visualize dose diffusion and measure the dose diffusion coefficient in VAM resin for the first time using a two-step printing procedure. We then directly image the projector's PSF to obtain the optical contribution. Conveniently, both the effects of diffusion and optical projection by a finite PSF can be expressed by simple 3D convolutions of the intended target geometry with the diffusion kernel and projector PSF. We show that we can recover uniform polymerization time for a range of disk thicknesses that is not possible otherwise by deconvolving the target geometry by the known diffusion kernel and projector PSF. Moreover, we show that this result is

generalizable to complex 3D objects by printing an object with fine interconnected walls and pores, and another with fine gear teeth. These advancements in the fundamental understanding of VAM printing and in VAM printing capability propel VAM to the forefront of next-generation ultra-rapid fabrication techniques[8–10], opening up a wide range of applications from complex fluidic components to in-space manufacturing.

## Results

### Feature-size dependence of print time

In VAM, objects tend to polymerize over a short, but finite time window. To avoid uneven curing and deformations due to sedimentation, all regions of the print should begin to polymerize at the same time. However, in practice, significant spatial variation in the polymerization onset time is observed. Print time variation can be due to a number of factors, such as the method used to calculate projections[4] and inclusion or omission of absorption in the projection calculation[1]. Recently, we qualitatively observed systematic print time variations via OST imaging[3]: large features polymerize first, followed by small features. In this work, we quantify this observation with a test structure comprised of a stack of disks of varying thickness, as shown in Fig. 1a. We printed this test structure with a VAM printer in two different resins (DUDMA,

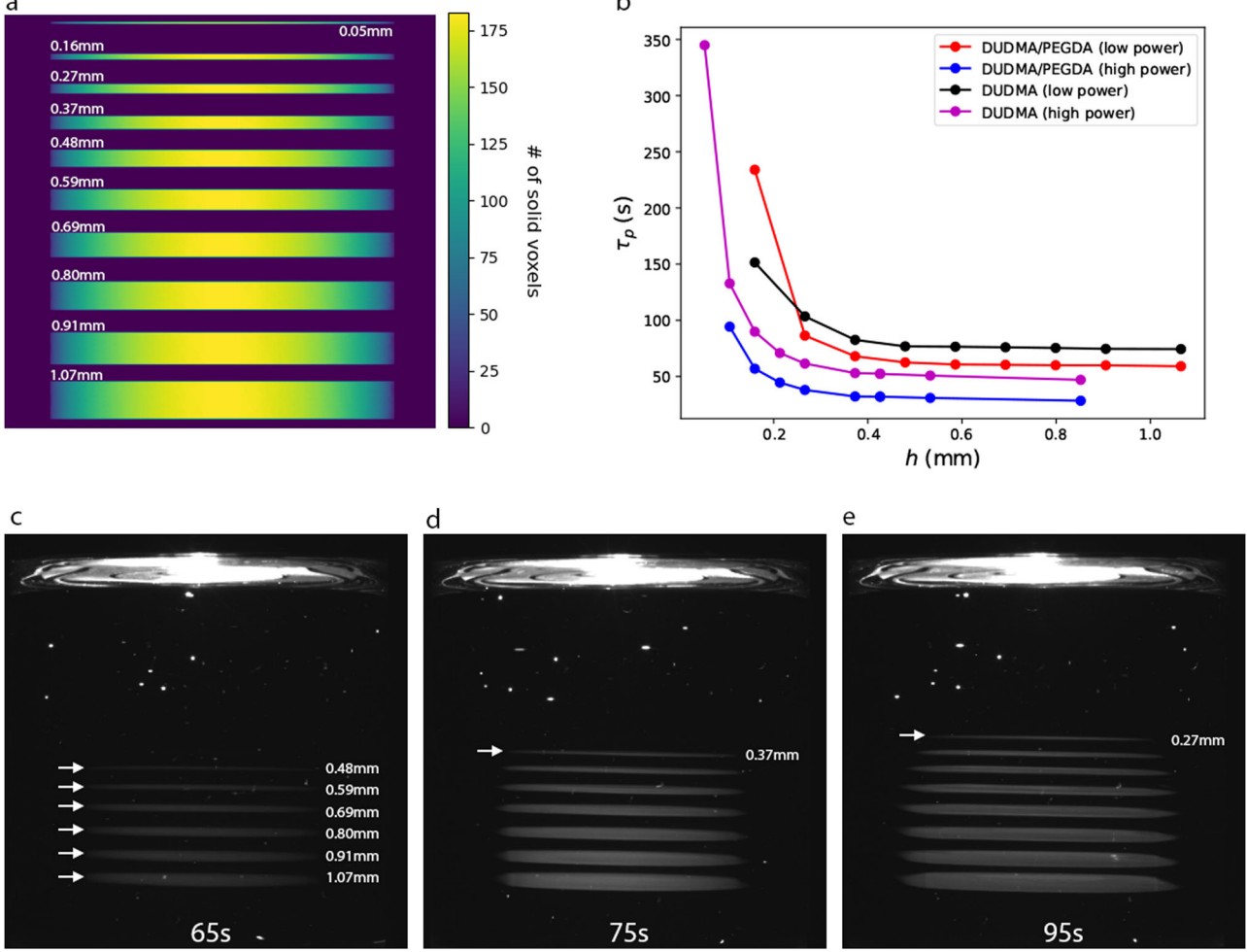

**Fig. 1 | Small features print slowly in VAM. a** Sum projection along the x-axis for a stack of disks with diameter 10 mm, with increasing thickness ($h$) from top to bottom (thicknesses as noted next to each disk). **b** Experimentally measured polymerization time ($\tau_p$) dependence on disk thickness ($h$) for DUDMA/PEGDA and DUDMA resins at low and high power (4 experiments total). Each datapoint is an individual experimental measurement. Datapoints of the same color are all obtained from a single print. **c–e** Raw experimental scattering images for a DUDMA/PEGDA resin print at low power, for timepoints 65 s, 75 s and 95 s. Thick disks polymerize first, followed by thin disks at later times.

$\mu = 8645$ cP and DUDMA/PEGDA, $\mu = 1750$ cP) each at two projector power settings (low power: 4.9 mW/cm$^2$; high power: 9.8 mW/cm$^2$), and recorded the time to polymerization, $\tau_p$ (Fig. 1b, See Methods–Print time measurement). In all cases, $\tau_p$ is constant for disks thicker than ≈0.5 mm, and then rises dramatically for thinner disks. This phenomenon can be observed directly via optical scattering imaging as shown in Fig. 1c–e. At 65 s (Fig. 1c), the thinnest polymerized disk is 0.48 mm. Thinner disks (0.37 mm and 0.27 mm) eventually polymerize with more exposure (Fig. 1d, e for exposure times of 75 s, 95 s, respectively).

## Diffusion in VAM

In VAM, light dose is applied over a period of approximately 10–100 s. During this period, oxygen is depleted; when it drops locally below a threshold concentration, polymerization is initiated[7,11–13]. While light-induced local oxygen depletion is occurring, oxygen will also diffuse back into the illuminated oxygen-depleted region. For fine features, oxygen will quickly diffuse back throughout the entire illuminated region, effectively slowing printing and increasing time to polymerization[7,14,15]. For features larger than the diffusion length, oxygen does not diffuse appreciably into the bulk of the illuminated region, and therefore has little effect on the time to polymerization. Note that in some VAM systems, the role of oxygen is replaced by a

radical quencher such as TEMPO[16,17]. VAM requires dissolved oxygen or a radical quencher to achieve dose contrast because even regions that are not intended to be cured receive light dose due to the geometry of tomographic light projection. In this section we model this diffusive effect and subsequently demonstrate it experimentally.

Below we outline our experiment for measuring the rate of this diffusive effect in VAM. We describe the diffusive phenomena in the resin via what we call the "dose diffusion coefficient" $D$. Instead of describing the mobility of a physical molecule in the resin, it describes the abstract concept of the diffusive spread (in mm$^2$/s) of the absorbed light dose in the resin. This mathematical treatment in terms of the light dose allows for a more convenient combination with light dose spread due to the projector PSF as described in the following two sections. Also, although we expect that oxygen is the most mobile diffusing species in the resin, larger species such as the photoinitiator are also (though much less) mobile[15]. Despite this description, we note that the underlying physical diffusion is still mediated by molecules in the resin, and dominated by the diffusion of oxygen.

To measure the dose diffusion coefficient, we performed the following experiment (Fig. 2a). First, we measure the exposure time needed to initiate polymerization of a $h_0 = 0.5$ mm thick disk. This threshold exposure time is measured to be $t_{th} = \{51.5(+0.4, -0.6)s, 73.5(+0.8, -2.7)s\}$ for DUDMA/PEGDA and DUDMA resins,

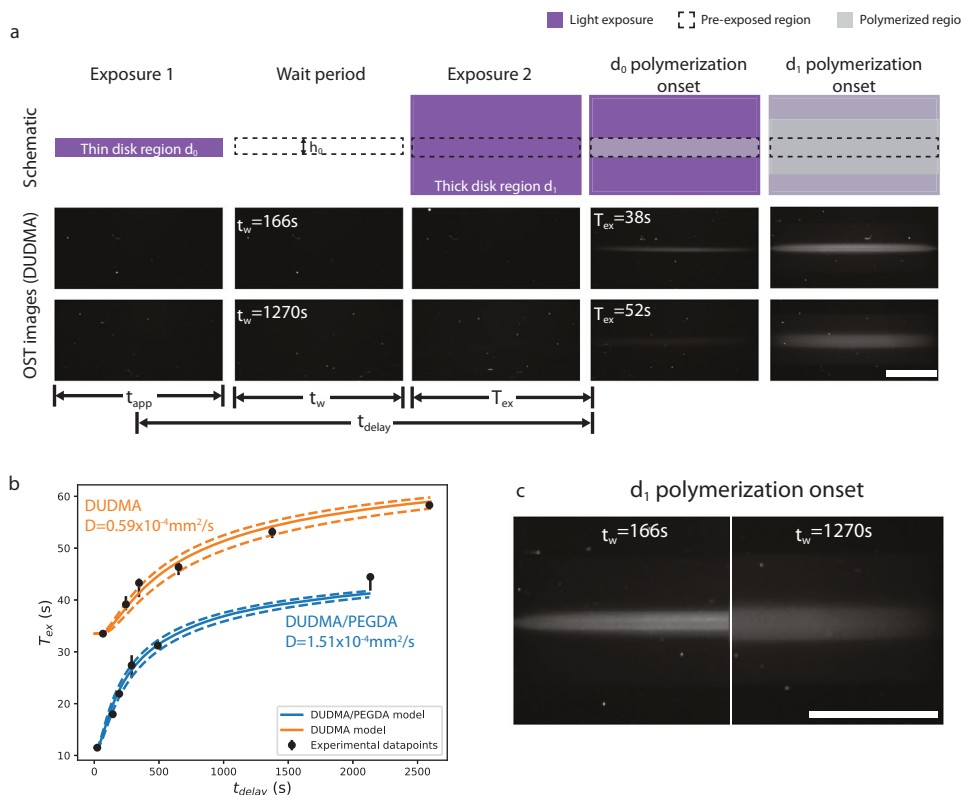

**Fig. 2 | Measurement of dose diffusion coefficient. a** Top row is a schematic of the experiment. First, a thin disk region (thickness 0.5 mm) is exposed for duration $t_{app} = 40s$ (exposure 1). This is followed by a wait period of duration $t_w$, during which there is no light. The wait time is changed for each experiment in the series. After the wait period, a second exposure occurs encompassing a thick disk (5 mm) is applied. First, this extra dose elevates the dose in the thin disk region to polymerization ($d_0$ polymerization onset), after an extra time $T_{ex}$ of exposure. The total time over which diffusion occurs is approximated as $t_{delay}$ (see Eq. 1). As exposure continues, the entire thick disk ($d_1$ region) polymerizes. Raw OST images for each of these steps are shown in the middle and bottom rows for wait times of 166 s and 1270 s, respectively. For shorter wait times, a small extra exposure time is required to polymerize the thin disk. **b** Experimental data points for the diffusion

experiment showing the relationship between extra exposure time needed to polymerize the thin disk ($T_{ex}$) and the delay time $t_{delay}$. The upper and lower bounds of the error bars correspond to measurements of $T_{ex}$ when setting polymerization threshold at 3 and 1 gray levels above the background, respectively. The datapoints correspond to a polymerization threshold of 2 gray levels above the background. Each datapoint represents a separate individual experiment. The curves are fits to the data assuming dose diffusion (dotted curves indicate lower and upper bounds, see text). The resulting dose diffusion coefficients for each resin are noted next to teach curve. **c** Raw scattering images of the thin disk region at the onset of polymerization of the thick disk. For the longer wait time, the polymerized region is thicker due to diffusion. Scalebars are all 5 mm, but do not include magnification by the vial.

respectively. These values are measured by recording the first time point for which the scattering intensity exceeds the background value by 2 camera gray levels (camera noise level ≈ 1 gray level in an 8-bit image). Upper and lower error bounds correspond to intensities of 3 and 1 gray levels above background level, respectively.

Next, in a separate print, we repeat the projections for the $h_0 = 0.5$ mm thick disk but terminate exposure after $t_{app} = 40$s. We denote the region of this disk as $d_0$. This 40 s exposure is slightly below the threshold exposure times $t_{th}$ and so therefore the disk does not print. Oxygen is significantly depleted within $d_0$, but not enough for polymerization to occur. After this sub-threshold exposure, the vial remains in the dark for a specified wait time $t_w$. During this entire sequence, oxygen is diffusing back into the oxygen-depleted region $d_0$. After this delay, we expose a disk with thickness ($h_1 = 5$ mm) centered at the same point as the first disk. We denote the region of this second, thicker, disk as $d_1$. This second step depletes oxygen uniformly in the neighborhood surrounding the original depleted zone $d_0$. After this second exposure begins, the thin disk starts to polymerize when the oxygen concentration is decreased to below the polymerization threshold. Alternatively, we can say that the thin disk starts to polymerize when the dose concentration is increased to above the polymerization threshold. The total amount of time over which oxygen is diffusing into the thin disk region is approximated by the delay time:

$$t_{delay} = t_w + T_{ex} + t_{app}/2 \qquad (1)$$

Here, we make the approximation that instead of applying dose evenly throughout the initial period $t_{app}$, we instead apply the same amount of total dose instantaneously at time $t_{app}/2$. In Eq. 1, $T_{ex}$ is the extra time needed to initiate polymerization of the thin disk ($d_0$) after the wait period (Fig. 2a). $T_{ex}$ is related to $t_{delay}$, $D$, and the time to reach polymerization threshold at fixed mean projector intensity $t_{th}$ via (see Supplementary Information for derivation):

$$T_{ex} = t_{th} - t_{app} \times \mathrm{erf}\left(h_0/4\sqrt{Dt_{delay}}\right) \qquad (2)$$

where erf is the Error Function. The extra exposure time required to initiate polymerization increases with $t_{delay}$. This is because although the total dose is conserved, it diffuses to occupy a larger volume, therefore decreasing its peak value. From Eq. 2 we can determine the $D$ by recording the extra exposure time needed to initiate polymerization of the thin disk, $T_{ex}$, for a series of $t_w$ values, each resulting in experimentally measured $t_{delay}$ values. $T_{ex}$ values are recorded with the same intensity threshold criteria as described above for $t_{th}$. The associated error bars in Fig. 2b correspond to threshold intensities as described above for $t_{th}$.

In Fig. 2b, we show experimental $T_{ex}$ vs. $t_{delay}$ plots for two resins with differing viscosities, and corresponding fits to the data using the model in Eq. 2. We find excellent agreement between the fits and the data, yielding $D = 1.51(+0.16, -0.20) \times 10^{-4}$ mm$^2$/s for DUDMA/PEGDA and $D = 0.59(+0.08, -0.10) \times 10^{-4}$ mm$^2$/s for DUDMA. The upper and lower error bounds correspond to fit results obtained using the upper and lower bounds measured for $t_{th}$ and $T_{ex}$ as described above. The effect of the diffusion can be readily observed when comparing polymerized regions for different $t_w$ values, as shown in Fig. 2c. Larger $t_w$ (1270 s) allows for more diffusion, which in turn leads to a thicker polymerized region (Fig. 2c, right) than with a smaller $t_w$ (166 s, Fig. 2c, left).

For a typical print time of $t_{print} = 60$ s, these diffusion coefficients indicate a three-dimensional dose diffusion length of up to $l_d = \sqrt{6Dt_{print}} \approx 0.15 - 0.23$ mm for dose applied at the beginning of the print. Although dose applied later in the print does not diffuse for as long and therefore remains more localized, the diffusion of the dose applied at the beginning of the print is responsible for a significant increase in print time for features smaller than the diffusion length.

## Optical point spread function

In addition to dose diffusion, optical effects also spread-out dose application. For an ideal projector, each square pixel of the projector chip (in our case, a digital micromirror device, DMD) is projected into the resin as a square with perfectly sharp edges. However, infinitely sharp edges are not possible in a real projection system due to diffraction and finite étendue of the light source. In practice, the intensity profile of the image of a projector pixel in the resin is spread out beyond the size of the pixel. We will call the average intensity profile of a single pixel in the write volume of the vial the optical point spread function (PSF) of the system, $\mathrm{PSF}_{opt}(\mathbf{r})$. The actual dose applied to the resin (before diffusion) can be calculated by a convolution of the ideal projected dose $A(\mathbf{r})$ and $\mathrm{PSF}_{opt}(\mathbf{r})$.

We directly measure the optical PSF of our VAM system by imaging fluorescence emitted by the resin when illuminated with a line. Supplementary Fig. S1 shows both overhead and side profile images of fluorescence arising from lines projected into the resin. To calculate an average PSF width in the print volume, we fit a 1D Gaussian to the average intensity profile of the projected line across the 10 mm diameter of the disk test structure from Fig. 1. In the xy and yz planes, the best fit Gaussian FWHMs are $0.120 \pm 0.004$ mm and $0.190 \pm 0.004$ mm, respectively, where the errors are one standard deviation of the fit results given by the SciPy optimize.curve_fit function. The difference in FWHM is due to the astigmatism imparted by the curved vial in our non-index-matched VAM printer.

The spread of the dose due to the optical PSF of the system has a similar effect to diffusion, except that the optical PSF is time-independent. Increasing the viscosity or reducing print time has no effect on the dose spread caused by a PSF of finite size.

## Combined diffusion and optical PSF model

Both oxygen diffusion and the optical PSF need to be considered together to explain the increase in exposure time needed to polymerize small features. Conveniently, the effective 3D projected dose in the resin can be written as a convolution of the target object light dose $A(\vec{r})$ with the optical PSF ($PSF_{opt}$) and a diffusion kernel $D_k$:

$$I_{proj} = A * \left(\mathrm{PSF}_{opt} * D_k\right) = A * \mathrm{PSF}_{eff} \qquad (3)$$

The first convolution is a result from incoherent imaging theory[18], and the latter from the fundamental solution to the diffusion (or heat) equation[19].

In Eq. 3 we assume $\mathrm{PSF}_{opt}$ is a 3D Gaussian with FWHMs measured in the previous section. For $D_k$, we must consider that dose is applied throughout the duration of the print exposure. The diffusion kernel for dose projected at the beginning of the print will have a larger width than for dose applied at the end of the print. For simplicity, we approximate the effect of diffusion throughout the print by summing a single 3D diffusion kernel for each rotation of the vial:

$$D_k(\mathbf{r}) = \sum_{n=0}^{N-1} e^{-|\mathbf{r}|^2/(4D(n+1/2)\Delta t)} / \left(4\pi D(n+1/2)\Delta t\right)^{\frac{3}{2}} \qquad (4)$$

Where $D$ is the dose diffusion coefficient, $n$ indicates the rotation number, $N$ is the total number of vial rotations during the print, and $\Delta t$ is the period of vial rotation. Supplementary Fig. S2 shows 1D slices of $\mathrm{PSF}_{eff}, \mathrm{PSF}_{opt}$, and $D_k$ for our printer using two different viscosity resins (DUDMA/PEGDA and DUDMA). Although diffusion plays a smaller role in the higher viscosity DUDMA resin compared to the DUDMA/PEGDA resin, it must be included to fully account for the feature size dependence of polymerization time. This can be observed in Fig. 3 (two experiments per resin) and Fig. S5 (3 replicates for one resin) where we plot the experimental polymerization time for disks of varying thickness, $h$, along with the polymerization time predicted by the combined optical PSF and diffusion model above (dashed curves,

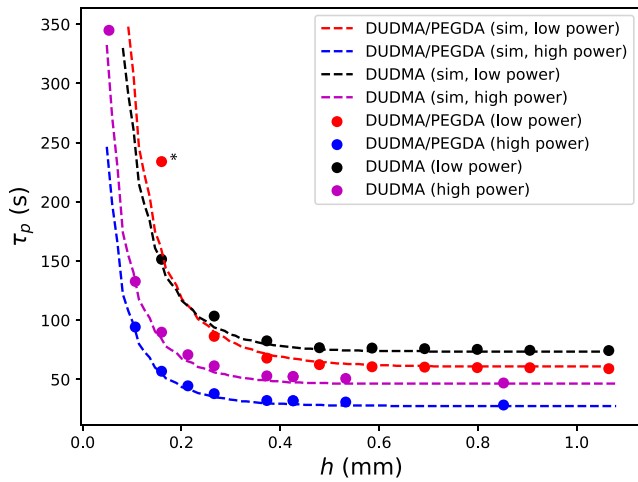

**Fig. 3 | Simulations based on diffusion-optical dose spread in the resin accurately describe print time variation with disk thickness.** Polymerization time $\tau_p$ for disk thickness $h$ ranging from 0.1 mm–1 mm. Experimental data are plotted as circles, with datasets using both low- (4.9 mW/cm²) and high-power (9.8 mW/cm²) printing illumination for each resin (total 4 experiments). Each datapoint is an individual experimental measurement. Datapoints of the same color are all obtained from a single print. The dashed curves correspond to simulations (sims) using the diffusion coefficients and optical PSF widths found in the previous section. We observe very close agreement between the model and the experimental data, using the diffusion coefficients and optical PSFs measured above, for both resins and for two different illumination levels. These curves are not fits to the data, which underscores the accuracy of the model. If either diffusive or optical PSF effects are removed from the model, the resulting curves do not match our experimental observations (Fig. S3).

The datapoint in Fig. 3 for the $h = 0.16$ mm disk in DUDMA/PEGDA at low power (marked by *) deviates from the model prediction. Further inspection reveals that this disk does not fully form – the disk quickly deforms, tilts and begins to rise in the resin (Fig. S4). This behavior is likely due to convective flow from heat generated by the thicker cured disks below. As such, this datapoint is not expected to be captured by simulation, which does not account for any heat transfer or flow phenomena. The disruption of polymerization of small features by flow further underscores the importance of synchronizing polymerization times across the print.

**Correction**
Ideally, all features in a VAM print will polymerize at the same time. If some features appear before others, the print becomes susceptible to a variety of deformations. First, regions that polymerize more rapidly will start to sediment in the resin due to the density difference between the liquid monomer and solid polymer. Second, the polymerized resin has a higher refractive index compared to the liquid precursor, and therefore will cause distortions in the projected light dose which may cause further errors in the print. Third, and most crucially, regions that are slow to polymerize require extra light dose. This necessitates increased exposure time for the entire print, which overexposes regions that polymerized first. This causes large errors in objects with both big (> 0.5 mm) and small (< 0.5 mm) features; either the small features are exposed until polymerization occurs with large features being overexposed, or the large features are correctly exposed, in which case small features are underexposed and do not appear. In a typical VAM workflow, small features are often underexposed, and their absence is only discovered in the print verification stage.

In this section we describe a deconvolution method for correcting for non-simultaneous polymerization in VAM. In Fig. 3 we established that the feature size dependence of polymerization time is accurately modelled via the effect of a diffusion kernel, $D_k$, and an optical PSF, $PSF_{opt}$. The result is that the target projected dose is low pass filtered by a convolution with $PSF_{eff}$, as shown in Eq. 3.

To determine the required projector patterns that more closely match the desired projected dose, we can mitigate the effect of the diffusive and optical low pass filter via deconvolution. Mathematically, we are seeking to find a deconvolved target object light dose $A_{dc}$ that yields the original target object light dose $A$ after convolution with the effective PSF ($PSF_{eff}$), yielding a projected dose $I_{proj,dc}$ that is the same as the original object $A$:

$$I_{proj,dc} = A_{dc} * PSF_{eff} = A \tag{5}$$

To find an approximate solution for $A_{dc}$, we use a modified Richardson-Lucy (RL) optimization algorithm. In imaging applications, RL deconvolution is often used to restore images degraded by non-ideal optics and has the advantage of avoiding negative values in the output[20]. Here, we have a similar goal of restoring a high-quality projected dose.

Our deconvolution algorithm is as follows (available as Supplementary Code 1). We start with a standard RL iteration (Eq. 6), followed by an intensity normalization step (Eq. 7).

$$A_{dc}^{(n_i+1)} = A_{dc}^{(n_i)} \cdot \left( \frac{A}{A_{dc}^{(n_i)} * PSF_{eff}} * PSF_{eff} \right) \tag{6}$$

$$A_{dc}^{(n_i+1)} = A_{dc}^{(n_i)} \cdot \frac{A}{A_{dc}^{(n_i)} * PSF_{eff}} \tag{7}$$

The standard RL step provides a sharpening effect, while the intensity normalization step ensures that the projected dose of small features is the same as for large features. We alternate applying Eqs. 6 and 7 for a small number of iterations (typically until $n_i = 2 - 10$). Note that deconvolution is performed on the entire 3D volume. After we obtain $A_{dc}$, we calculate projection images from $A_{dc}$ (instead of $A$) in the standard way for tomographic VAM (see Methods–Projection calculation). Supplementary Fig. S5 shows an example of $A_{dc}$ and $A$ in the case of the disk geometry of Fig. 1, as well as the effective delivered doses. As expected, the deconvolution procedure results in a more uniform effective dose for all disks, regardless of thickness (Fig. S5b, d).

In imaging applications, deconvolution is used to approximately reverse the effect of the low pass filter *after* image acquisition due to optical system performance or motion blur. For our application, we instead need to deconvolve the image *before* projection. We note that other deconvolution methods are possible, such as division by the effective PSF in Fourier space[6]. We favor the RL approach above as it guarantees that the result is non-negative, as required for physical light projection. Although deconvolution can precompensate for suppression of high spatial frequencies in the projected dose, it cannot enable projection of spatial frequencies beyond the passband of the projections optics, nor can it alleviate dose spread due to vial wobble or convective flow.

**Correction results**
Next, we tested deconvolution correction using our VAM printer for DUDMA and DUDMA/PEGDA resins using the same stacked disk geometry as in Figs. 1 and 3. The results are plotted in Fig. 4 as the relative polymerization time $\tau_{p,rel} = \tau_p(h)/\tau_p(h = 1.07 \text{ mm})$. This allows us to compare the relative performance of the correction methods

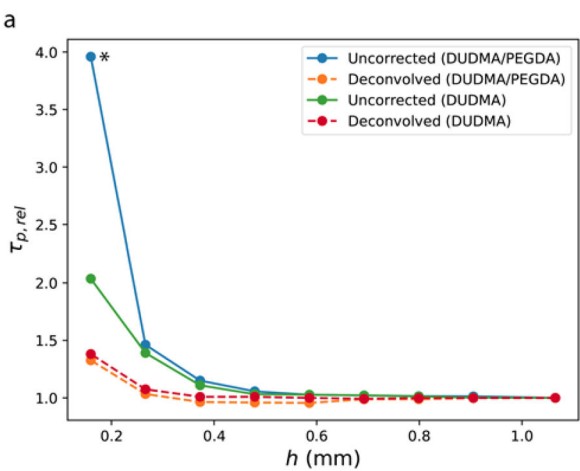

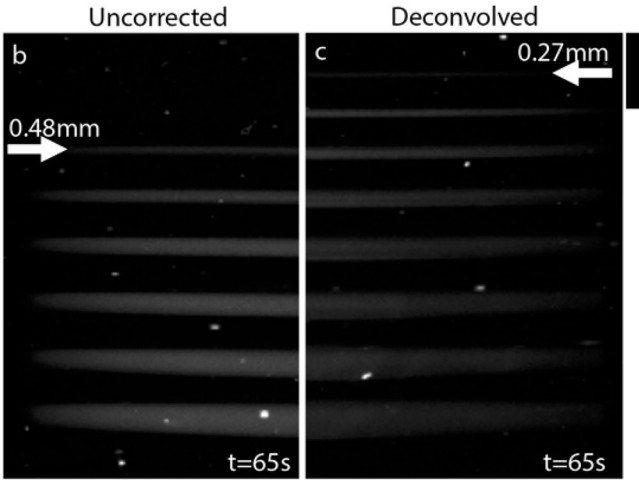

**Fig. 4 | Deconvolution pre-corrects for dose spreading, yielding near-simultaneous printing for all disk thicknesses. a** Experimentally measured relative polymerization time $\tau_{p,rel}$ as a function of disk thickness $h$ for uncorrected (solid) and deconvolved (dashed) projections. Each datapoint is an individual experimental measurement. Datapoints of the same color are all obtained from a single print. Deconvolved projections provide a significant improvement in the simultaneity of the prints for $h$ below approximately 0.5 mm. Data shown are for the low power projector setting (4.9 mW/cm$^2$). The y-axis is plotted in logarithmic scale for clarity. (*) Indicates disk that does not form properly, likely due to thermal effects. **b** Raw optical scattering image for an uncorrected disk print. The arrow indicates the thinnest formed disk at the t = 65 s ($\tau_{p,rel}$ =1.18). **c** As in **b** but for deconvolved projections. Here, t = 65 s corresponds to $\tau_{p,rel}$ =1.13. Scale bar is 2 mm in the vertical direction.

irrespective of the absolute print time. An ideal correction method should yield $\tau_{p,rel}$ =1 over as large of a size range as possible.

As expected, deconvolution provides a significant improvement in the simultaneity of disk polymerization times in comparison to uncorrected prints. For disks with $h \geq 0.27$ mm printed in DUDMA/PEGDA resin, the maximum difference in polymerization times is 7.7% when using deconvolution compared to 46.2% for the uncorrected print. For the thinnest disk ($h$ = 0.16 mm) the difference in relative polymerization time is extreme: in the uncorrected case, this disk requires a 296% larger dose than the thickest disk, compared to 33% more for the deconvolution corrected print. Though a significant improvement, this thin disk still requires a longer exposure to reach polymerization. We suspect that this may be due to the limited dynamic range of the projector (8-bit). More importantly, the disk forms properly, without the distortion observed for the uncorrected print (Fig. S4).

For the more viscous DUDMA resin, we observe a maximum difference in polymerization times of 8.5% for $h \geq 0.27$ mm. Again, this is significantly improved over the uncorrected case where there is a 39.1% difference in polymerization times between $h$ = 0.27 mm and $h$ = 1.07 mm disks. The difference is even more pronounced for the $h$ = 0.16 mm disk, which requires 104% more dose than the $h$ = 1.07 mm disk in the uncorrected case, compared to 39.1% in the deconvolved case. These results can be qualitatively appreciated by observing raw optical scattering images for DUDMA/PEGDA prints as shown in Fig. 4b, c. For the uncorrected print, disks $h \geq 0.48$ mm have formed at $t$ = 65 s (Fig. 4b), compared to disks $h \geq 0.27$ mm at the same time point for the corrected print (Fig. 4c).

Deconvolution correction can have a dramatic effect for objects where small features are critical to the overall structure. In such a case, it can be impossible to print small features before large features overexpose completely and start to sediment. Such a situation is shown in Fig. 5, where we attempt to print a cylindrical gyroid structure with variable wall thickness. This gyroid has a variable porosity that is minimum at the top and bottom and maximum in the middle. The geometry of this object (designed with nTopology, USA) is shown in Fig. 5a–c (see Fig. S9 for detail). The gyroid unit cell is 3 mm, with a variable wall thickness maximum at the top (≈1 mm), leaving only small holes leading into the interior of the gyroid pore space (Fig. 5a,

Fig. S9d). In the middle, the minimum wall thickness is 0.18 mm, leaving only thin sinusoidal features (Fig. 5c, Fig. S9e). We expect that the thick top and bottom regions should appear well before the middle and may overexpose due to how long the middle must be exposed to reach the polymerization threshold. This is confirmed by optical scattering images of the print volume that clearly show the top and bottom of the cylindrical gyroid appearing first beginning at 54.5 s (top left, Fig. 5d). The middle of the cylindrical gyroid gradually starts to form, and the top and bottom appear to just start to join at 78 s (bottom left, Fig. 5d), at which point the projection exposure is stopped. However, at this time, the top and bottom of the cylindrical gyroid are significantly overexposed, leading to an hourglass shape. Moreover, the part has begun to sediment, compromising the alignment between the top and bottom parts of the object. The result is a failed print that does not resemble the intended cylindrical gyroid geometry (see Fig. 5d at 87.0 s).

Optical scattering images of the print volume for a deconvolution-corrected print are shown in Fig. 5e. In this case, the entire cylindrical gyroid forms at the same time (≈96 s). Due to this simultaneity, there is no overexposure of top and bottom compared to the thinner middle region. The intended print geometry is faithfully reproduced despite its small features.

In Fig. 6 we show cross-sectional views of the printed and as-designed variable wall thickness gyroid from Fig. 5. The OST isosurface of the uncorrected print in Fig. 6a shows that the internal pore structure is completely lost due to the nonuniform polymerization time across the structure. In contrast, Fig. 6b confirms that the complex interior geometry of the gyroid has been faithfully reproduced in the deconvolution-corrected print. A detailed inspection of a series of 2D cross sections shows that the size of both small negative and positive features appears correctly when using deconvolution correction, but not without (Fig. S9b, c, f–i). Remarkably, the interior pore space of the deconvolved gyroid print is also intact, indicating that both fine positive and negative features (the pores) are formed. This is key for fluidic applications, where the fidelity of negative features across a range of sizes is crucial.

Photographs of the printed objects in Fig. 6g, h further evidence how deconvolution correction is critical to achieving a successful print. Via optical microscopy, we directly confirmed that the diameter

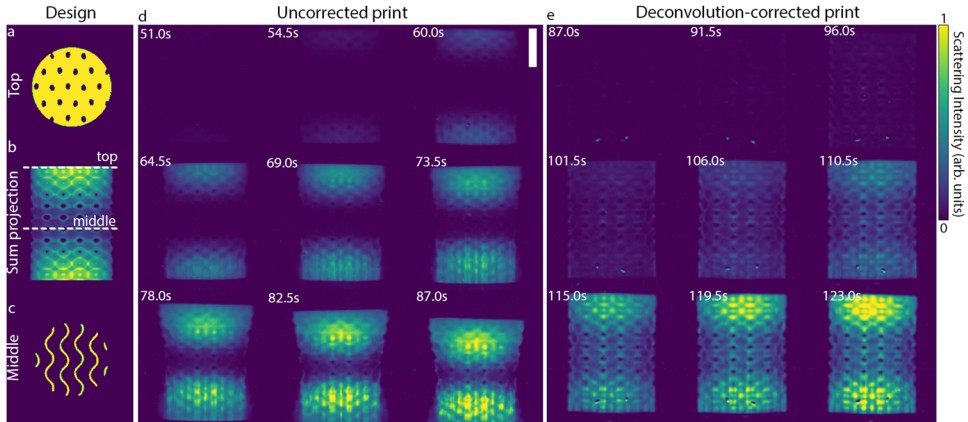

**Fig. 5 | Deconvolution enables VAM printing of a structure with a large range of feature sizes. a** The top layer, **b** sum projection, and **c** middle layer of the design file for the variable wall thickness gyroid. The horizontal axis is scaled by 1.5x to match the scaling of optical scattering images in (**d**) and (**e**). **d** Optical scattering images of the build volume for an uncorrected gyroid object with varying wall thickness. Gyroid walls are thickest at the top and bottom (1.31 mm), and thinnest in the middle (0.18 mm). The top and bottom of the gyroid form first at 54.5 s, followed by the middle at ~78 s, at which point the top and bottom are overexposed and have started to sediment, resulting in print failure. Scalebar is 5 mm vertically (3.33 mm horizontally due to vial refraction). **e** Same as in **d** but for a deconvolved gyroid volume. Here, the entire gyroid polymerizes at the same time (~96 s). The print forms correctly without overexposure at the top and bottom.

of a top surface pore (580 μm, Fig. 6i) is close to the design diameter (≈ 500 μm, Fig. S9d). Furthermore, we measured the thinnest walls in the printed part to be in the range of 195–200 μm by cutting the gyroid in the middle and imaging with an optical microscope (Fig. 6j). This value is slightly larger than the design value of ≈180 μm.

Using OST data, we also compare the maximum wall thickness of the deconvolution-corrected print along its height, with the uncorrected print and the design file (Fig. S10). We find that due to the overcuring of the uncorrected print at the top and bottom, the wall thickness of the uncorrected print far exceeds that of design file (RMS error 1.09 mm). In contrast, the deconvolution-corrected print follows the variation in the design file very closely, with 8x lower RMS error (RMS error 0.131 mm). This increase in print quality is also quantitatively captured by the 3D Jaccard similarity index[4] (see Methods), which increases from 0.44 for the uncorrected print to 0.69 for the deconvolution-corrected print.

Another example of print improvement via deconvolution correction is shown in Fig. 7. Here, we fabricated a pair of mechanical gears in the same print (Fig. 7a). The top gear has wide teeth (0.4 mm), whereas the bottom gear has thin teeth (0.1 mm). As expected, there is a large difference in polymerization time of the thick and thin teeth of the top and bottom gears, as can be seen in optical scattering images (Fig. S11). This leads to significant overcuring of the top gear (Fig. 7b), which is necessary to drive the teeth of the bottom gear to polymerization. The bottom gear is also overcured due to the larger feature size of the gear ring compared with the teeth.

Deconvolution correction yielded significantly improved print fidelity as shown in Fig. 7c. All teeth and central hole formed in both gears and are not overcured. The 2D Jaccard similarity (see Methods) of the front face of the gear increased for both gears (0.69 (corrected) vs. 0.62 (uncorrected) for the gear with thin teeth, 0.79 (corrected) vs. 0.69 (uncorrected) for the gear with wide teeth). For both gears, deconvolution correction achieves a lower RMS error for teeth width and length, as summarized in Table S1. Like the variable wall thickness gyroid in Figs. 5, 6, the gears printed in Fig. 7 are an example of a geometry with a wide range of feature sizes that is not possible to print using VAM without deconvolution correction.

## Discussion
In this work we systematically investigated a common phenomenon in VAM—that finer features require a larger light dose to polymerize[7]. We explain our observations using a model that includes time-dependent

dose diffusion and time-independent dose spread due to the optical PSF of the projected pattern in the resin; we measure the magnitude of both contributions directly.

Previous VAM papers have estimated oxygen diffusion (which mediates the local effective dose) coefficients of $\sim 10^{-7} mm^2/s$ in a 10,000 cp resin[2], $\sim 10^{-6} mm^2/s$ in 100,000 cp resin[4], and $\sim 10^{-3} mm^2/s$ in 12 cp resin[7]. These estimates are based on comparison to values reported in the literature but were not confirmed experimentally. In this work, we experimentally measure a proxy for oxygen diffusion - dose diffusion - in situ for the first time in VAM, obtaining diffusion coefficients on the order of $0.5 - 1.5 \times 10^{-4} mm^2/s$ for resin viscosities of 1750 cp (DUDMA/PEGDA) and 8645 cp (DUDMA). Our quantitative results are complemented by clear qualitative visualization in Fig. 2c that the polymerized region expands as the wait time between two exposures is increased. That we are able to observe this difference in our imaging setup without microscale resolution shows that the diffusion length is far greater than previously estimated in some systems (e.g. ~ 30 μm vs. ~ 2 μm over 20 s[2]).

These dose diffusion coefficients alone do not fully account for the size-dependent polymerization times measured in Fig. 1. We found that the inclusion of the optical PSF in the resin into the model in Eq. 4 admits good fits to the data over a range of resin viscosities and intensities, indicating that this model generalizes well to a broad range of print parameters.

We believe that the dose diffusion-optical PSF blurring model in Eq. 4 likely explains many subtle artifacts in our and other groups' previous VAM works[2-7], as described in the introduction. Furthermore, diffusion and/or optical blurring are also known to be present in other printing techniques, such as light-sheet 3D microprinting[21], two-photon 3D printing[14,15], and continuous liquid interface production (CLIP) printing[9].

Size-dependent polymerization time can be mitigated by reducing dose diffusion via increased viscosity, or faster print times via higher light intensity or photoinitiator concentration. Likewise, high resolution projection optics with large depth of field will also serve to reduce this adverse effect. Both fundamental and practical restrictions on these physical mitigation techniques, however, make software pre-correction via deconvolution an attractive solution that is applicable irrespective of the resin and projector optics.

In this work, we demonstrated computational correction of this size-dependent polymerization time by deconvolving the intended object volume prior to projection calculation (Figs. 4–7). We highlight

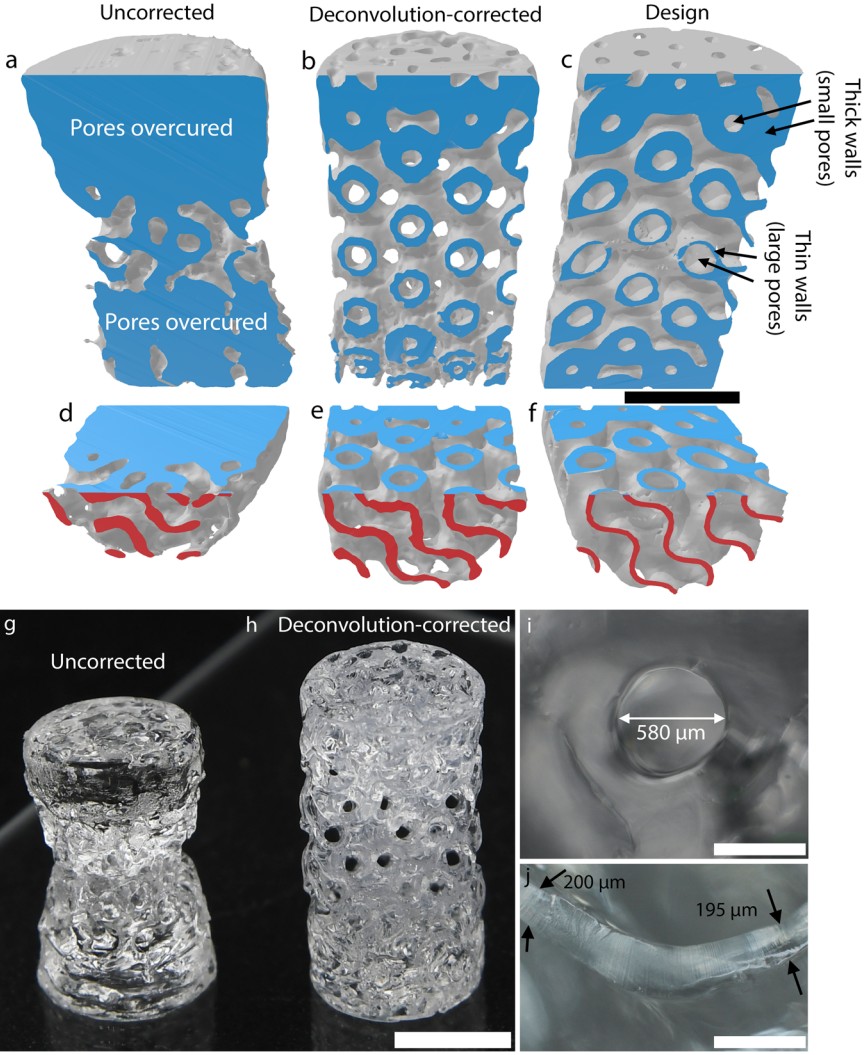

**Fig. 6 | Deconvolution yields high fidelity final part where uncorrected VAM fails due to large range of feature sizes. a** OST isosurface cross-section for an uncorrected print of the variable wall thickness gyroid from Fig. 5. **b** OST isosurface cross-section for the deconvolution-corrected print. OST Reconstruction quality at the bottom of gyroid is slightly compromised due to residual dirt on glass vial. **c** Cross-section of the variable wall thickness gyroid for reference. **d**–**f** As in

**a**–**c**, with an additional horizontal cross section along the midplane shown (face color in red). **g** Photograph of the uncorrected print. Photograph of the deconvolution-corrected print. **i** Optical microscope image of a pore on the top surface of the deconvolution-corrected gyroid. **j** Optical microscope image of an interior wall of the gyroid after cutting horizontally. Scale bars for (**a**–**h**) are 5 mm. Scalebars for (**i**) and (**j**) are 0.5 mm.

the effectiveness of this approach by printing a gyroid object with highly variable features sizes and a complex internal pore structure. This object was previously impossible to print using VAM, yet deconvolution yields a remarkably high-fidelity result (Figs. 5–6). Likewise, we demonstrated fabrication of mechanical parts with microscale features (Fig. 7) that are not printable without deconvolution-correction. This advancement opens the door to fabricating functional objects with variable feature sizes such as high frequency RF antennas[22,23], microfluidics[24] and thermal management devices[25], as well as hierarchically architected materials[26–28] using VAM.

We expect that deconvolution correction will be of particular importance for low viscosity resins like those typically used in vat photopolymerization, where diffusion effects are more pronounced. Initial experiments indicate that deconvolution correction may provide benefit for VAM printing down to viscosities of ~100cp (Fig. S12), with further investigation planned.

As a rule of thumb, for typical resins and VAM PSF widths, deconvolution is necessary for features sizes less than 0.5 mm. The exception to this is for lattice structures where the feature size is small but nearly uniform throughout. In this case, there is no harm in

exposing longer to reach polymerization for small lattice features, as dose spread will be approximately uniform throughout, resulting in an overall delay in polymerization. This likely explains why deconvolution correction was not needed to fabricate lattices with small struts (50 μm) in recent VAM work[16].

The print exposure time in VAM is typically decided subjectively by the operator based on real time video[3] (as was done in this work), or is pre-timed based on resin properties. Computer automated exposure methods are a current area of research in VAM, however, if the polymerization time varies across the object, pre-timed or computer-vision based approaches will not be able to identify the correct time to terminate print exposure. By ensuring uniform polymerization time for the entire object, deconvolution correction creates the printing conditions necessary for future automated operation of VAM printers.

Recently, deconvolution of the design object was used to pre-compensate for dose blurring in VAM printing of scattering resins, yielding an improvement in the fidelity for scattering objects[6]. Our deconvolution method is complementary as it includes not only static contribution of light transport in the resin but also the dynamic process of dose diffusion. The latter is critical to proper correction as we

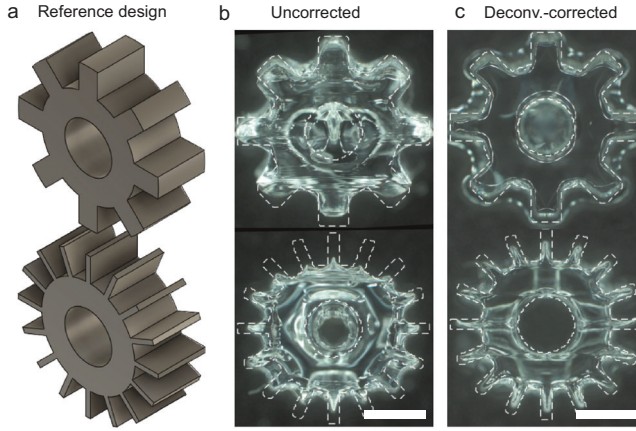

**a** Reference design **b** Uncorrected **c** Deconv.-corrected

**Fig. 7 | Printing a gear with VAM, enabled by deconvolution correction.**
**a** Isometric view of the reference design of two gears with different tooth thickness.
**b** The mechanical gears in **a**) printed without deconvolution correction. **c** The
mechanical gears in **a**) printed with deconvolution correction. The white-dashed
lines in **b**, **c**) represent the boundary of the reference design. All scale bars are 1 mm.

demonstrate in Fig. S3, where diffusion is ignored. While our full
combined model does not consider the projector's depth of field or
depth-dependent scattering blur, it does include the anisotropy of the
projector's PSF. In future work, a full software-based correction might
combine all these effects into a single deconvolution framework.

All AM techniques are prone to modality-specific artifacts, for
example: stringing in fused deposition modeling (FDM)[29], undesired
porosity in metal powder bed fusion[30], balling in selective laser
melting[31] and micro-explosions in two-photon polymerization[32,33].
Since VAM's inception, many limitations of VAM have been lifted,
including widening print materials to include scattering resins[6],
ceramics[34] and glass[16]; more accurate projection calculation
algorithms[4,35], stiffness control[36], striation removal[37], simpler opto-
mechanical design[5,38] and live visualization[3]. Maturing VAM into a
mainstream commercial-grade technology will require investigation
and correction of VAM-specific challenges such as the feature size
discrepancy addressed in this paper. Our research significantly
improves the capabilities of VAM, enabling the production of high-
quality prints with a wide range of feature sizes without modifying the
printer or incurring additional hardware costs.

## Methods

### Resins
Diurethane dimethacrylate (DUDMA, Esstech Inc.) was mixed with
1.75 mM ethyl (2,4,6-trimethylbenzoyl) phenylphosphinate (TPO-L,
Oakwood Chemical) to prepare the DUDMA resin. DUDMA/PEGDA
resin was composed of 80 wt% DUDMA, 20 wt% poly(ethylene glycol)
diacrylate (Mn = 700 g/mol, PEGDA 700, Sigma Aldrich), and 1.75 mM
TPO-L. PEGDA700 resin for Fig. 7 was prepared with 3.38 mM TPO-L.
The mixed photoresins were kept refrigerated and protected from
light until use. The average viscosities of DUDMA, DUDMA/PEGDA and
PEGDA 700 resins are 8,645 cP, 1,750 cP, and 100 cP respectively,
measured using a TA Instruments Discovery Hybrid Rheometer 2 at
25 °C. The refractive index of the resins was measured using a Schmidt-
Haensh ATR-P refractometer. At 405 nm, the refractive indices of
DUDMA, DUDMA/PEGDA and PEGDA 700 resins are 1.503, 1.500, and
1.485 respectively.

### Polymerization time simulations
The dashed curves in Fig. 3 and solid and dashed curves in S5-S7 were
obtained by convolving a unit dose square function (the dose pro-
jected to form the disk) by the diffusion kernel and the PSF for each
time step. The dose needed to initiate polymerization is found by

recording the dose obtained at the (experimentally measured) time of
polymerization onset for the thickest disk. The simulated poly-
merization onset time curve is then found by looping through in time
and recording at the time at which the dose for a disk thickness
exceeds the dose required for polymerization. We perform this cal-
culation for 100 equally-spaced disk thicknesses ($h$ values) between
0.054 mm and 1.08 mm. The diffusion kernel is updated for each time
step (0.5 s) up to a maximum of 375 s. This fine time step yields a
denser temporal sampling than would be obtained by only updating
every vial rotation, which is how we calculate the PSF for deconvolu-
tion purposes. The curves shown in Fig. 3 terminate at the smallest
writable disk size within the simulation time of 375 s. In practice,
convective flow from the exothermic polymerization reaction and vial
wobble further limits the thinnest printable disk. Simulations were all
written in Python.

### Projection calculations
Projection calculations were performed in a similar manner as pre-
viously reported[3]. Briefly, a STL CAD model is imported into Python
and voxellized using the Trimesh package[39]. Solid voxels have value 1,
empty (background) voxels have value 0. For the bolt/nut (Fig. S12)
print, background voxels were set to 0.25. As previously reported[3,35],
this helps to alleviate artifacts from subsequent ramp filtering (see
next paragraph). Next, deconvolution correction was applied via
Eqs. 6–7. Note that deconvolution is applied in 3D, not layer-by-layer.

The resulting deconvolved volume is then sliced into layers. Each
layer is Radon transformed and then ramp filtered. Negative pixel
values are set to 0. The resulting ramp filtered Radon transforms are
reassembled into a sequence of 2D images, one image for each angular
sample. Calculations are performed at an angular sampling rate of 1
image per degree. Finally, projections are corrected to counteract the
lensing effect of the vial and the non-telecentricity of the projector, as
in refs. 3,5.

### Printing
For Figs. 1–6 and S1–10, printing was performed with a VAM printer
similar to that previously reported in ref. 3, except that in this work, the
projector is outfitted with a 405 nm LED light source. The rotation rate
used was 40°/s. The projected pixel pitch in the vial was 0.054 mm
(vertical) and 0.037 mm (horizontal). The horizontal pixel pitch is
reduced by a factor of $n_{resin} \approx 1.5$ compared to the vertical pixel pitch
due to lensing of the vial. The camera used to capture optical scat-
tering data for polymerization time studies and OST capture is a FLIR
Grasshopper USB 3 GS3-U3-23S6M-C running in 8-bit mode.

For Figs. 5–7, prints were terminated when the smallest features
were visible to the operator in the live optical light scattering video.
The variation of these prints due to subjectivity in the print termina-
tion time (< 5 s) is minimal compared to the difference in poly-
merization time of large and small features for uncorrected prints
(20 s). For Figs. 5, 6, the smallest features are the walls at the middle of
the gyroid.

For Figs. 7 and S11, printing was performed with a VAM printer
described in a previous work[38]. The rotation rate used was 60°/s, and
the projected pixel pitch in the vial was 0.011 mm (vertical) and
0.007 mm (horizontal). Again, the reduction in horizontal pixel pitch
relative to the vertical is due to the cylindrical lensing of the vial.

The parts in Fig. S12 were printed using the spaceCAL printer
onboard a microgravity parabolic flight. Details on this printer can be
found in the supplementary information "Microgravity printer details
(SpaceCAL)" and in ref. 40. The resin was contained in a cylindrical
glass tube with end caps on both sides to avoid leakage while in
microgravity. First, an endcap was attached to one side of an empty
glass tube. A volume of resin was poured into the tube, such that when
the other endcap was attached the entire volume was resin without any
air pockets. Prints were exposed for a timed 20 s window

corresponding to the parabolic flight maneuver. For this print, the mean intensity was increased scaling the intensity so that 85% of all pixel values were within 2 standard deviations from the max pixel value. This was required to print within the allotted 20 s time window. The rotation rate used was 54°/s.

## Post processing

For all prints, objects were transferred out of the vial with a metal spatula and left to soak in isopropyl alcohol (IPA) for 10 min. After the IPA soak, prints were left to dry at room temperature. Prints in shown in Fig. 6 and S9 were then post cured in a Formlabs Form Cure for 60 min at room temperature.

Prints in Figs. 7, S11, and S12 were placed in a vacuum chamber and continuously pumped with 405 nm flood illumination for 15 min to post cure.

## Print time measurement

The print time for the disk experiments in Figs. 1, 3, 4 was set to the first time point at which a given disk appeared in the optical scattering image, with a brightness 50% above the background. In Figs. S5, S7, S8, we plot data from 3 replicates of uncorrected and deconvolution-corrected (Figs. S7, S8) prints in DUDMA/PEGDA resin. In each set of datapoints, two replicates are performed in immediate succession using the same batch of resin, and the third replicate is from a different batch of resin, printed on a different day.

## Visualization

OST isosurface reconstructions were first saved as STL files via a Python script and then viewed in Microsoft 3D Builder. For the cross sections in Fig. 6a–f, the intersection of the isosurface model and the cross-sectional planes are colored blue and red for visibility.

## Jaccard similarity index

The 3D Jaccard similarity index for the gyroid prints in Figs. 5, 6 were computed in Python. First, we aligned the 3D volume generated using OST with the 3D design file (both rotation and translation). Both the OST and design file are 3-dimensional NumPy arrays with value 0 outside the solid part and 1 inside. The intersection and union of these volumes are then calculated using matrix algebra in NumPy. The Jaccard similarity index is then calculated as the intersection divided by the union.

The 2D Jaccard similarity index for the parts shown in Fig. 7 were calculated in a similar procedure as above but using a microscope image of the top face of the part (Fig. 7) and a 2D slice of the 3D design file. For each microscope image, the solid region of the gear was determined based on the edge of the part within the depth of field of the microscope.

## Gear teeth geometry measurement

The length and width of each gear tooth in the 4 objects in Fig. 7 were measured and compared to their respective design values in Table S1. Tooth width measurement was performed by manually identifying the edges of teeth midway from base to tip and measuring this width using the ruler tool in Adobe Photoshop. Tooth length was measured by manually identifying the distance between base to tip and measuring this distance with the ruler tool.

## Data availability

All data needed to evaluate the conclusions in the paper are present in the paper and/or the Supplementary Materials. Source data are provided with this paper. Raw OST data are available upon request. Source data are provided with this paper.

## Code availability

Sample Python code for object deconvolution is provided as a Supplementary Code file.

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

## Acknowledgements

The authors would like to thank Kurtis Laqua and Derek Aranguren van Egmond for stimulating discussion on this topic. This work was funded in part by the National Research Council Canada Ideation Fund (project NBR3-627-1) (A.O.), and by the NASA Suborbital Tech Flight program, award number 80NSSC21K0350 (H.K.T.).

## Author contributions

Conceptualization: A.O., D.W., Y.Z., C.P., H.W.D.H., H.K.T., T.W., J.B., Methodology: A.O., D.W., T.W., Investigation: A.O., D.W., Y.Z., K.S., C.P., T.W., Th.L., Resin preparation: D.S., R.L., S.D., Y.Z., Th.L., K.S., Printing: A.O., D.W., Y.Z., Ta. L., Visualization: A.O., D.W., Software: A.O., D.W., Writing: A.O., D.W., Y.Z., C.P., J.B., H.K.T., H.W.D.H., K.S., Supervision: H.K.T., C.P., J.B.

## Competing interests

The NRC-affiliated authors have filed a US provisional patent relating to the correction method described in this paper (63/411719, applicant: National Research Council Canada). The authors declare no other competing interests.
