## [Peer Review File · Nature Communications]

Deconvolution volumetric additive manufacturingREVIEWER COMMENTS

Reviewer #1 (Remarks to the Author):

the authors show that by taking diffusion of an inhibiting species (like O₂) as well as the PSF of the projector in VAM printing, that they were able to correct for the polymerization time difference between different voxel sizes.

This is indeed an important effect that has been experimentally observed in VAM but not yet studied in detail, which is what this paper contributes.

The study is well executed and well written and is an important result.

The authors should clarify the meaning of this "diffusion coefficient" D when it is introduced first in the text. What is the physical meaning and the units? The term $D \cdot t_{\text{delay}}$ should be unitless according to eq. 2, but D is given in mm^2/s in the measurements.
line 131: " T_{rex} is related to t_{delay} , D , and the ..."

The intensity of the light scattered (projected along the plane of the disc) is used to estimate the thickness of the disc. It seems to me that ground truth (i.e. the true thickness) might not be linearly related to the amount of light reaching the camera because of multiple scattering i.e. when light is scattered by a thin disk, the high angles are lost (not captured by the camera) whereas when the thickness is larger, the light scattered by the same thin disk volume can be rescattered to the camera, hence producing more light than if the thin disk was not surrounded. Is this an effect that under appreciates the size of thin discs?

in line 195, define "perfect" PSF.

I do not understand the phrase "In reality the two differ slightly due to the positivity constraint". My understanding of your "target light dose" is already the physically realizable positively constraint light dose.

In Fig 5Sb, the delivered dose has a height thicker than the intended polymerized thickness. This indicates that there is a limit between adjacent discs which is given by the delivered dose thickness rather than the thickness of the polymerized part. In other words, is that correct that VAM can fabricate 0.16 mm thick discs but they cannot be made 0.16 mm apart?

I am not sure I understand fig 5. The pore size seems to be the same across the height (seen in 6b), so why would there be a difference in height. I must be missing something.

It would be worth mentioning that when printing with a solid material (such as GelMa), a gyrus can be well printed with similar performance as the corrected print shown by the authors. See fig 6 iii), xi), ix) in <https://doi.org/10.1002/adma.202110054>

The part on the microgravity is not very convincing. The diameter of the screw should be the same for corrected and uncorrected since the diameter is much larger than 0.5 mm and thus the difference should come in the height of the thread. It is indeed shown that the corrected have large pitch to valley but both should fit the nut...

Reviewer #2 (Remarks to the Author):

The paper might contain some new and significant scientific information adequate to justify publication, but in its current form, I recommend rejecting the submission but encouraging resubmission because of the following points:

The paper is interesting and relevant, in my opinion, but the structure and the reporting of the study need to be improved. The reporting of the experimental methods and results should be more complete and accurate. The main aim and focus of the paper should be sharpened – now there are

corrections, zero-gravity printing etc.

The abstract requires more details of the research and results – now it is at a very general level and the value of information is quite low.

The literature part should be widened to focus on the state of art. For example, open findings of “A brief literature review indicates that this is indeed a general phenomenon in VAM 2–7”

Results have parts that should be in the introduction, materials and methods or discussion. The structure of the manuscript is unclear and makes it very difficult to follow and understand. The materials and methods chapters are good in structure but are a missing example, the micro-gravity test, even though some details are scattered in M&M.

I recommend using example tables for the test parts made with parameters, materials etc.

In AM, I recommend sticking ISO/ASTM 52900:2021 Additive manufacturing — General principles — Fundamentals and vocabulary. For the sake of clarity and for future understandability and indexing when standard name overrules other. For example, all processes belong to the vat photopolymerization process category.

Explain in detail how the experiments of the samples have been planned and analyzed (factors, levels, type of experimental plan, replications, analysis of variance and related statistical tests. Why do the corrected parts not look much better than the non-corrected ones? Did you perform any measurements for 3D-printed corrected and non-corrected parts?

Explain how the resin will stay in the vat in the micro-gravity test and what phenomena are related to that.

About sample geometries – more details are required for 3D models or drawings with dimensions. Please revise the typos and language of the manuscript.

After these modifications, scientific content could be more deeply analyzed and reviewed.

Reviewer #3 (Remarks to the Author):

In this paper, the authors discuss a computational correction method to improve the quality of parts produced via volumetric additive manufacturing (VAM) techniques. Their findings suggest VAM as an increasingly mature printing method, capable of producing functional geometries in a more reliable fashion than previous work has achieved. Overall, the work seems relevant to ongoing efforts in AM, with printed parts demonstrating notable improvement from the inclusion of deconvolution. After reading the paper, my specific comments are as follows:

At various points in the manuscript, the authors mention the importance of “big” and “small” features in a single print (e.g., line 244). In the author’s estimation, what is the size range for each of these? Because of the large variation in build volumes across different AM processes, what is considered a “big” feature on one printer may be viewed as a “small” feature on another.

The print stopping criteria discussed in line 523 (...terminated when the smallest features were visible to the operator...) seem rather subjective. Was the operator the same for all prints? How did they know whether the feature counted as “visible”? This is especially of interest since, as the authors themselves mention in line 320, “small errors in small features often go unnoticed.” If minor variations in the end stopping point are not cause for concern in the final analysis, this should be discussed and explained by the authors.

While the authors qualitatively discuss the final quality of parts in Figure 6 and 6S, quantitative evidence is minimal, outside of some nominal measurements. It would significantly improve the results if more quantitative evidence was included. For example, is there a way to quantify the percent difference between the design intent, the uncorrected design, and the corrected designs? This concern also carries over to the discussion of the case study in Fig. 7.

The application of the work to the microgravity context is unclear. Though the high-quality build geometry is noted when compared with the build defects in the Earth environment, why investigate the role of microgravity at all? If the deconvolution approach cannot produce functional quality in Earth gravity, the authors need a stronger rationale for why microgravity environments in particular benefit from the use of VAM, not necessarily why VAM benefits from microgravity.

Reviewer #4 (Remarks to the Author):

Suggested: Major Revision.

I think this is a very interesting paper that clearly contributes to the literature on AM. Overall, I think this is a very promising paper, but it requires serious improvement to be published.

In General

1. The structure of this paper requires serious improvement. Please follow the standard paper structure based on IMRAD. This is an important issue of this study.

Abstract

The abstract is well-written and informative.

Introduction

1. The introduction doesn't fully articulate the research question or grasp the reader's interest.
2. A literature review has to be presented in this section. This is totally lacking. Please elaborate.
3. After presenting the state of the art, proceed with highlighting the gaps this study is addressing and describe the added value of your work.
4. After correcting the structure of the study as per IMRAD, include a short paragraph at the end of the introduction briefly describing the sections of the study. Consider adding a graphical abstract as it facilitates the readers' understanding and it will improve the visibility of your study.

Results

1. The approach/methods section is missing; it is not explained if any methodological approach has been considered. It is not acceptable to present the results after the introduction, without mentioning the approach you have used to obtain them.
2. Please provide details of the experimental equipment used for the presented experiments, as well as the design of the experiments. You can use a table to organize these.

Diffusion in VAM

1. The name of this section is not proper. Please improve the structure as per comment 1. Additionally, consider numbering the sections.
2. Some parts of this section are largely descriptive and they would be better moved in the introduction. Please point out the challenges you aim to overcome in the introduction.
3. "This threshold exposure time is... respectively": Please use a reference if this is taken from a source or describe the experimental setup and procedure used for this measurement.
4. Provide a description of the methods you use in a cohesive manner adding an approach section. The use of schematics is also encouraged, to facilitate the readers' understanding and improve the visibility of your study.
5. Equation 1 and wherever it appears: Within equations, numbers, punctuation, parentheses, common function names, units, or mathematical signs are set upright; variables are set in italics, and vectors are set in bold. Please refer to ISO80000-1. Please make the necessary corrections in all equations.
6. Fig.2: The caption of figure 12 is over ten lines. Please describe the figures in the main text of the manuscript and keep the captions short.

Optical point spread function

1. As per comment 8. This applies to all section titles of this study.
2. Eq.(4): Consider using stacked fractions for large equations.
3. Please correct any language-related mistakes like this (line 216): "These curves are not fits to the data".
4. Line 218: "Fig.S3". There is no such figure. Please correct this for the rest mentions of the figures as well.

Correction

1. As per comment 8. This applies to all section titles of this study.
2. Line 244-248: Please elaborate further on how you concluded that smaller features require more exposure time or provide a reference.
3. Lines: 273-275: Please consider adding a flowchart-schematic of the described steps.

Correction results

1. Provide details of the experimental setup as well as the design of the experiments used.
2. The term "Benchy print" is not formal. Consider using "benchmark test artifacts" or a similar term.
3. Line 327: "designed with nTopology, USA" please provide a reference.
4. Fig.5: The caption of figure 5 is over 5 lines. Please describe the figures in the main text of the manuscript and keep the captions short.
5. The correction you propose seems interesting. Please provide more detailed measurements and comparisons between the four different kinds of parts: microgravity-, microgravity corrected-, earth-, and earth corrected-parts. Additionally, a comparison of the manufactured parts with the part design should take place, to point out the maturity of this AM process.

Conclusions

1. Many pages in this section are not conclusions; a literature review is also included in the conclusions section. Please move those in the introduction. Please refer to comment 1.
2. Do not summarize the entire paper in this section, but rather the results and the discussion. Also, give some hints for future work.
3. After the conclusions, the acknowledgments, references, appendices, etc. should follow, and then the paper should come to an end. However, in this case, more sections follow, namely "Methods". Please correct this. Please refer to comment 1.

Methods

1. The section "Approach" or "Method" describes the theoretical framework that the paper suggests and does not necessarily refer to a specific case. An implementation and/or experimental setup should include such specified details.
2. You mention that simulations took place. Please elaborate on the simulation approach and software used.
3. Consider organizing the experimental data you provide in the "Printing" subsection in a table.

References

The manuscript could benefit from the following studies in terms of completeness of the conducted research:

- Foteinopoulos, P., Papacharalampopoulos, A., & Stavropoulos, P. (2018). On thermal modeling of Additive Manufacturing processes. *CIRP Journal of Manufacturing Science and Technology*, 20, 66-83.
- Salvekar, A. V., Nasir, F. H. B. A., Chen, Y. H., Maiti, S., Ranjan, V. D., Chen, H. M., ... & Huang, W. M. (2022). Rapid Volumetric Additive Manufacturing in Solid State: A Demonstration to Produce Water-Content-Dependent Cooling/Heating/Water-Responsive Shape Memory Hydrogels. *3D Printing and Additive Manufacturing*.
- Stavropoulos, P., & Foteinopoulos, P. (2018). Modelling of additive manufacturing processes: a review and classification. *Manufacturing Review*, 5, 2.
- Shusteff, M., Browar, A. E., Kelly, B. E., Henriksson, J., Weisgraber, T. H., Panas, R. M., ... & Spadaccini, C. M. (2017). One-step volumetric additive manufacturing of complex polymer structures. *Science advances*, 3(12), eaao5496.
- Stavropoulos, P., Foteinopoulos, P., & Papacharalampopoulos, A. (2021). On the impact of additive manufacturing processes complexity on modelling. *Applied Sciences*, 11(16), 7743.
- Salvekar, A. V., Nasir, F. H. B. A., Chen, Y. H., Maiti, S., Ranjan, V. D., Chen, H. M., ... & Huang, W. M. (2022). Rapid Volumetric Additive Manufacturing in Solid State: A Demonstration to Produce Water-Content-Dependent Cooling/Heating/Water-Responsive Shape Memory Hydrogels. *3D Printing and Additive Manufacturing*.
- De Beer, M. P., Van Der Laan, H. L., Cole, M. A., Whelan, R. J., Burns, M. A., & Scott, T. F. (2019). Rapid, continuous additive manufacturing by volumetric polymerization inhibition patterning. *Science advances*, 5(1), eaau8723.
- Obaton, A. F., Lê, M. Q., Prezza, V., Marlot, D., Delvart, P., Huskic, A., ... & Cayron, C. (2018). Investigation of new volumetric non-destructive techniques to characterise additive manufacturing parts. *Welding in the World*, 62(5), 1049-1057.

We thank the Editor and Reviewers for their comments. Our responses and changes to the main text are in red.

Revision note: there was a small error in the scale of the image in Fig. S1a, which subsequently propagated into the PSF_{xy} FWHM value (=0.08mm). This error has been fixed, leading to a corrected value for PSF_{xy} FWHM = 0.120mm. This only affected the print in Fig. 5,6. Because of this, this object was reprinted for this revision. All images, OST visualizations and derived quality metrics are from this updated print with the correct PSF_{xy} FWHM value. The other deconvolution parameters (D , PSF_z) remain unchanged; the net result on the print was therefore minimal.

REVIEWER COMMENTS

Reviewer #1 (Remarks to the Author):

the authors show that by taking diffusion of an inhibiting species (like O₂) as well as the PSF of the projector in VAM printing, that they were able to correct for the polymerization time difference between different voxel sizes. This is indeed an important effect that has been experimentally observed in VAM but not yet studied in detail, which is what this paper contributes. The study is well executed and well written and is an important result.

- The authors should clarify the meaning of this “diffusion coefficient” D when it is introduced first in the text. What is the physical meaning and the units? The term $D \cdot t_{\text{delay}}$ should be unitless according to eq. 2, but D is given in mm^2/s in the measurements. line 131: “ T_{rex} is related to t_{delay} , D , and the ...”

In Eq. 2, $\sqrt{Dt_{\text{delay}}}$ must have the same units as the numerator, h_0 (mm). Indeed, with D having units of mm^2/s , then $h_0/\sqrt{Dt_{\text{delay}}}$ is unitless as required as the argument of the error function.

An intuitive understanding of diffusion coefficient is that it is the square of the average displacement of a quantity (in this case the dose or oxygen molecule) per unit time.

The reviewer may have missed the introduction of D in the “Diffusion in VAM subsection of Results”. We have added the units of D in this explanation to make it clear that this D has the same units as any normal diffusion coefficient (line 107).

- The intensity of the light scattered (projected along the plane of the disc) is used to estimate the thickness of the disc. It seems to me that ground truth (i.e. the true thickness) might not be linearly related to the amount of light reaching the camera because of multiple scattering i.e. when light is scattered by a thin disk, the high angles are lost (not captured by the camera) whereas when the thickness is larger, the light scattered by the same thin disk volume can be rescattered to the camera, hence producing more light than if the thin disk was not surrounded. Is this an effect that under appreciates the size of thin discs?

The reviewer raises a good point. If there is multiple scattering, this would hamper the quantitative nature of the OST technique. However, we have shown in a previous paper that OST is fully quantitative and does not suffer from this issue (ref 3). This is supported qualitatively by the observation that these resins are very transparent throughout the printing process – there is very little internal scattering observable by the naked eye.

In the case when the illuminating OST light is linearly polarized, the side scattered light collected by the camera has a high degree of polarization. This indicates that the side scattered light is dominated by single scattering events that do not scramble polarization.

Below we have included a pair of images of a Thinker print in DUDMA/PEGDA resin under linearly polarized illumination, and with a linear (detection) polarizer in front of the camera. When the detection polarizer is aligned with the illumination polarization, the side scattered image is bright. When the detection polarizer is turned 90° , the scattering intensity drops by a factor of $\sim 60\times$, indicating that multiple scattering events that scramble the polarization are very rare. Note that the right-hand image (perpendicular detection polarizer) is taken with $10\times$ the integration time compared to the image with the parallel polarizer (left). The degree of linear polarization within the region of the print is $\sim 96-97\%$. This is only possible when there is a very small amount of multiple scattering. We include these images for the benefit of the review process; however, we have not added them to the manuscript as they do not related directly to this work.

Fig. 1. Left: Thinker print illuminated with linearly polarized light (horizontal direction of the image) and imaged through a polarizer oriented horizontally (same direction as illumination polarization). Camera integration time: 50ms. Right: As in left, but with detection polarizer oriented at 90° to illumination. Camera integration time: 500ms.

We note that we are not measuring the thickness of the disks to make any quantitative assertions in this paper. Rather, we measure the time to onset of polymerization to measure the diffusion coefficient. This requires only observing when scattering intensity starts to increase, but it does not require any measurement of the disk geometry.

- in line 195, define “perfect” PSF.

This has been removed – see response to next question.

- I do not understand the phrase “In reality the two differ slightly due to the positivity constraint”. My understanding of your “target light dose” is already the physically realizable positively constraint light dose.

A target light dose with zero dose outside the part is not physically realizable because we cannot project “negative” intensity. This is what we intended to address with this sentence. However, upon further reading this is confusing and not necessary, so we have removed it from the text.

- In Fig 5Sb, the delivered dose has a height thicker than the intended polymerized thickness. This indicates that there is a limit between adjacent discs which is given by the delivered dose thickness rather than the thickness of the polymerized part. In other words, is that correct that VAM can fabricate 0.16 mm thick discs but they cannot be made 0.16 mm apart ?

The reviewer is correct in that the delivered dose is thicker than the intended thickness, which is due to both the optical PSF and dose diffusion, as outlined in this work. In general, small positive features (disks) are easier to print than small negative features (spaces between disks) – a common observation in 3d printing and optical lithography. To make thin disks, one can always underexpose the resin so that only the middle of the disks cure. This is possible because the polymerization threshold of the resin introduces a nonlinearity into the printing process. Because of this nonlinearity, the disks can in principle be made arbitrarily close to each other by allowing the exposure to continue until the distance between the edges of two adjacent cured disks reaches 0.16mm. However, achieving this in practice is challenging, and we do not claim to have achieved this level of fidelity in this work.

- I am not sure I understand fig 5. The pore size seems to be the same across the height (seen in 6b), so why would there be a difference in height. I must be missing something.

The internal structure of the variable wall thickness gyroid is difficult to visualize in 2D. We have now included Fig. S9a, which shows several cross sections along the height of the gyroid. Evidently, there are small pores at the top and bottom, compared to large open pore spaces in the middle region. The new Fig. 6 also helps to visualize the interior of the print. We have labeled the thick walls / small pores at the top vs. the thin walls / large pores in the middle.

We have kept Fig. 5a,c we have annotated the top and middle as thick walls and thin walls, respectively. Fig. 5b is a sum projection. The pores do not go straight through the gyroid, but rather oscillate back and forth. This is why the small pores near the top and bottom do not show up as holes in the sum projection. The value in showing the sum projection, however, is that the optical scattering image a sum projection of the scattering

density through the build volume. In the deconvolution corrected print, the optical scattering images in Fig. 5e match the sum projection in Fig. 5b very well.

Indeed, it is straightforward to print a standard gyroid structure in VAM (also see our previous work in ref 3). The difference here is that the variable wall thickness gyroid has a feature size (the wall thickness) that changes significantly along its length (~1mm to <200um). Hopefully Fig. S9a helps the reviewers and readers to understand the geometry and the resulting challenge with this demonstration. Compare the section at the top left of Fig. S9a (1.5mm from top) to the middle section (7.5mm from top). The top section is mostly thick walls punctuated by small holes. The middle section is nearly all empty space, except for thin sinusoidal features. In a normal gyroid, the thickness of the walls is uniform throughout, which is not the case in our variable wall thickness gyroid.

- It would be worth mentioning that when printing with a solid material (such as GelMa), a gyroid can be well printed with similar performance as the corrected print shown by the authors. See fig 6 iii), xi) , ix) in <https://doi.org/10.1002/adma.202110054>

We respectfully disagree with this assessment given the published data available in the paper linked above. The thick walls at the entrance and exit of the structures of Fig. 6 are much thicker in the printed objects compared to the design (design ~2mm, printed ~2.65mm in iv.), despite the overall shrinkage in the printed part vs. the design. The ratio of the outer diameter to inner diameter is ~3.5 in the design file but is increased to ~5 in the printed file. Whereas the design file is a cylinder in the above paper, the printed objects all take on a dumbbell shape. This is very similar to (though less pronounced than) our uncorrected print in Figs. 5-6. Furthermore, it is unclear whether the fine internal structure of the objects in Fig. 6 of the linked paper does indeed form properly (Fig. 6A iv and v do not and appear to be missing unit cells. vi does seem to, but do not have thin walls). We do not wish to speculate as to the origin of these inaccuracies as we do not have access to experimental details. However, we note that these departures from design are very similar to what prompted us investigate the phenomena in our paper. We emphasize that we have specifically demonstrated that both fine and large features of our prints form properly via OST and imaging the final parts using a light microscope.

We note that the paper linked above does not report the dose diffusion coefficient for GelMA, nor the PSF width of the projector, or the width of the interior structure walls, so it is not possible to know whether the range of feature sizes are all above the PSF size (and diffusion length) or not. If the thin walls of the internal structures are smaller than the PSF width, then we would expect that deconvolution correction would improve on the results in the above linked paper.

The part on the microgravity is not very convincing. The diameter of the screw should be the same for corrected and uncorrected since the diameter is much larger than 0.5 mm and thus the difference should come in the height of the thread. It is indeed shown that the corrected have large pitch to valley but both should fit the nut...

We have changed the outlook on the microgravity work and moved it to the supplementary information (Fig. S12). This dataset is valuable in that it shows that this approach may be crucially applicable to low viscosity resins where diffusion is much faster than current highly viscous VAM resins. Currently the best way to test this is on a parabolic flight to remove confounding gravitational effects. However, as the reviewers have pointed out, further experiments need to be performed to establish the repeatability of this result and eliminate other possible explanations of improved print fidelity with the deconvolution-corrected print. Given the long lead time of performing VAM on parabolic flights, further experiments are out of the scope of this revision. Nevertheless, we wish to keep a reference to this figure in the discussion section (lines 451-454) to motivate compelling future directions of our deconvolution approach.

Reviewer #2 (Remarks to the Author):

The paper might contain some new and significant scientific information adequate to justify publication, but in its current form, I recommend rejecting the submission but encouraging resubmission because of the following points:

The paper is interesting and relevant, in my opinion, but the structure and the reporting of the study need to be improved. The reporting of the experimental methods and results should be more complete and accurate. The main aim and focus of the paper should be sharpened – now there are corrections, zero-gravity printing etc.

- The abstract requires more details of the research and results – now it is at a very general level and the value of information is quite low.

The abstract has been rewritten to include more specifics about the results, to the extent possible within the Abstract word count limit.

- The literature part should be widened to focus on the state of art. For example, open findings of “A brief literature review indicates that this is indeed a general phenomenon in VAM 2–7”

We have now moved the more detailed description of this literature review from the discussion to the introduction (lines 44-51).

- Results have parts that should be in the introduction, materials and methods or discussion. The structure of the manuscript is unclear and makes it very difficult to follow and understand.

We have done our best to follow the style of Nature Communications manuscripts, which typically have a lot of introductory and discussion material in the Results section. We have been advised by the Editor to keep with the current organization of the paper. However, we welcome specific examples of content that would benefit from being moved to another section.

- The materials and methods chapters are good in structure but are a missing example, the micro-gravity test, even though some details are scattered in M&M.

More detail on the micro-gravity test has been added in the Supplementary Information under the Section titled: Microgravity printer details (SpaceCAL).

- I recommend using example tables for the test parts made with parameters, materials etc. In AM, I recommend sticking ISO/ASTM 52900:2021 Additive manufacturing — General principles — Fundamentals and vocabulary. For the sake of clarity and for future understandability and indexing when standard name overrules other. For example, all processes belong to the vat photopolymerization process category.

We have included a new table (Table S2), that summarizes each of the prints performed in this paper.

We have now included the term “porosity” to describe the variable wall thickness gyroid (line 332). Likewise, we also make use of the term vat photopolymerization in the text.

- Explain in detail how the experiments of the samples have been planned and analyzed (factors, levels, type of experimental plan, replications, analysis of variance and related statistical tests.

We have included three new figures (Figs. S5, S7, S8) that provide more statistical context for the stack of disk polymerization experiment in Fig. 1. We see a small variation in polymerization time over 3 replicates of both uncorrected and deconvolution-corrected disk polymerization experiments. Interestingly, the variation in polymerization time increases with decreasing disk thickness. We suspect this may be due to vial wobble, which can vary from vial to vial depending on how it sits in the vial holder and the intrinsic geometry of the vial. A description of how the replicate experiments were performed is provided in the Methods sections under “Polymerization time measurement”.

Experiment design details are all reported in the main text. These details are now summarized in the “Experiment design” section of the supplementary information in bullet point form.

- Why do the corrected parts not look much better than the non-corrected ones? Did you perform any measurements for 3D-printed corrected and non-corrected parts?

We respectfully disagree with this comment. To better draw the reader’s attention towards the improvement in print quality, we have modified Fig. 6. The leftmost OST model (Fig. 6a) shows the uncorrected print. This print has no internal pore network. The difference between this uncorrected print and the deconvolution-corrected print (Fig. 6b) is very clear. Fig. 6b has a complex internal pore network that is readily visible in cross section. Again, these internal pores are almost entirely a solid blob in the uncorrected print (Fig. 6a). We show microscope images of an isolated wall and pore of the deconvolution-corrected print in Fig. 6i,j, with annotated dimensions. These dimensions match very well with the design file.

We have also included a new dataset (Fig. 7, Fig. S11 and Table S1) that further demonstrates a case where deconvolution correction is vital. Here, a qualitative comparison is made using the outline of the design geometry. The improvement in print fidelity is readily apparent. We have also added Table S1 which summarizes the length and width of the gear teeth from Fig. 7 along with the corresponding RMS error compared to design. This table is referenced just above Fig. 7 in the main text.

To quantify the increase in print fidelity for both Fig. 6 and 7, we now report the Jaccard index, which is a standard metric used in VAM for reporting geometrical accuracy. In both cases deconvolution correction increases this number significantly. Furthermore, for Fig. 6, we also plot the wall thickness as a function of vertical position in Fig. S10. The deconvolution-corrected wall thickness curves match the design file extremely well; the uncorrected wall thickness does not. The RMS errors for these plots and the Jaccard index are reported in lines 376-383 of the main text.

- Explain how the resin will stay in the vat in the micro-gravity test and what phenomena are related to that.

We have added the following text to the Printing section in Methods: “The resin was contained in a cylindrical glass tube with end caps on either side. First, an endcap was attached to one side of an empty glass tube. A volume of resin was poured into the tube, such that when the other endcap was attached that the entire volume was resin without any air pockets.”

- About sample geometries – more details are required for 3D models or drawings with dimensions.

Please see new Figs. S9 and S10 for more details on the geometry of the gyroid. The geometry of the gears in Fig. 7 are described in the main text above the figure and are summarized in Table S1.

- Please revise the typos and language of the manuscript.

We have made our best effort to correct typos.

After these modifications, scientific content could be more deeply analyzed and reviewed.

Reviewer #3 (Remarks to the Author):

In this paper, the authors discuss a computational correction method to improve the quality of parts produced via volumetric additive manufacturing (VAM) techniques. Their findings suggest VAM as an increasingly mature printing method, capable of producing functional geometries in a more reliable fashion than previous work has achieved. Overall, the work seems relevant to ongoing efforts in AM, with printed parts demonstrating notable improvement from the inclusion of deconvolution. After reading the paper, my specific comments are as follows:

- At various points in the manuscript, the authors mention the importance of “big” and “small” features in a single print (e.g., line 244). In the author’s estimation, what is the size range for each of these? Because of the large variation in build volumes across different AM processes, what is considered a “big” feature on one printer may be viewed as a “small” feature on another.

We have updated this part of the text to read: “. This causes large errors in objects with both big ($> 0.5\text{mm}$) and small ($< 0.5\text{mm}$) features...” (lines 254-255)

- The print stopping criteria discussed in line 523 (...terminated when the smallest features were visible to the operator...) seem rather subjective. Was the operator the same for all prints? How did they know whether the feature counted as “visible”? This is especially of interest since, as the authors themselves mention in line 320, “small errors in small features often go unnoticed.” If minor variations in the end stopping point are not cause for concern in the final analysis, this should be discussed and explained by the authors.

The need to manually identify print completion is a challenge for the field of VAM. There is ongoing work in our group and others to address this problem, however, it is outside the scope of this paper.

We have added text regarding print time subjectivity in the Methods section (lines 532-537). For Figs. 5-7, the variation due to subjectivity (<5 sec) is far less than the variation due to feature size-driven polymerization time dependence (20 sec). Furthermore, operator-based termination uncertainty affects over/undercuring uniformly across all size scales in deconvolution-corrected prints, whereas it leads to systematic over/undercuring depending on the local feature size, leading to more distorted prints. Printing deconvolution-corrected geometries is therefore more tolerant of termination time errors than uncorrected objects.

We note that for any automated print termination approach to work with VAM, the termination time must be the same across the entire print, regardless of feature size. Therefore, this paper is a necessary advance towards the goal of having an automated approach to print timing in VAM, because it enables the entire print to polymerize at the same time. We have added some text to this effect at lines 454-459.

Operator 1 performed all prints except for Fig. 7 and Fig. S12. Both prints in Fig. 7 were done by Operator 2. The prints in Fig. S12 were pre-timed due to experimental constraints.

- While the authors qualitatively discuss the final quality of parts in Figure 6 and 6S, quantitative evidence is minimal, outside of some nominal measurements. It would significantly improve the results if more quantitative evidence was included. For example, is there a way to quantify the percent difference between the design intent, the uncorrected design, and the corrected designs? This concern also carries over to the discussion of the case study in Fig. 7.

We have removed Fig. S6 and Fig. 7 in favor of a new Fig. 7 in the main text. Figure S6 was not an appropriate print to demonstrate deconvolution correction due to almost all the features being above the 0.5mm threshold where deconvolution becomes necessary. The improvement in print quality in Fig. 7 is shown quantitatively in Table S1.

To quantify the increase in print fidelity for both Fig. 6 and 7, we now report the Jaccard index, which is a standard metric used in VAM for reporting geometrical accuracy. In both cases deconvolution correction increases the Jaccard index significantly (lines 381-382, 399-401). Furthermore, for Fig. 6, we also plot the maximum wall thickness as a function of vertical position in Fig. S10. The deconvolution-corrected wall thickness curves matches the design file extremely well; the uncorrected wall thickness does not. The RMS errors for these plots and the Jaccard index are reported in lines 367-373 of the main text.

- The application of the work to the microgravity context is unclear. Though the high-quality build geometry is noted when compared with the build defects in the Earth environment, why investigate the role of microgravity at all? If the deconvolution approach cannot produce functional quality in Earth gravity, the authors need a stronger rationale for why microgravity environments in particular benefit from the use of VAM, not necessarily why VAM benefits from microgravity.

Here we repeat our answer to a similar question raised by Reviewer #2:

We have changed the outlook on the microgravity work and moved it to the supplementary information (Fig. S12). This dataset is valuable in that it shows that this approach may be crucially applicable to low viscosity resins where diffusion is much faster than current VAM resins. Currently the best way to test this is on a parabolic flight to remove confounding gravitational effects. However, as the reviewers have pointed out, further experiments need to be performed to establish the repeatability of this result and eliminate other possible explanations of improved print fidelity with the deconvolution-corrected print. Given the long lead time of performing VAM on parabolic flights, further experiments are out of the scope of this revision. Nevertheless, we wish to keep a reference to this figure in the discussion section (lines 451-454) to motivate compelling future directions of our deconvolution approach.

Reviewer #4 (Remarks to the Author):

Suggested: Major Revision.

I think this is a very interesting paper that clearly contributes to the literature on AM. Overall, I think this is a very promising paper, but it requires serious improvement to be published.

In General

1. The structure of this paper requires serious improvement. Please follow the standard paper structure based on IMRAD. This is an important issue of this study.

The paper structure is based on the Nature Communications format, so it cannot be changed.

Abstract

The abstract is well-written and informative.

Introduction

1. The introduction doesn't fully articulate the research question or grasp the reader's interest.

We respectfully disagree, though we have altered the introduction to include a more significant literature review.

2. A literature review has to be presented in this section. This is totally lacking. Please elaborate.

A literature review has been moved from the discussion to the introduction.

3. After presenting the state of the art, proceed with highlighting the gaps this study is addressing and describe the added value of your work.

The gap (uneven polymerization time dependent on feature size) is identified in the introduction (lines 39-66).

4. After correcting the structure of the study as per IMRAD, include a short paragraph at the end of the introduction briefly describing the sections of the study. Consider adding a graphical abstract as it facilitates the readers' understanding and it will improve the visibility of your study.

This suggestion does not conform to the Nature Communications format.

Results

1. The approach/methods section is missing; it is not explained if any methodological approach has been considered. It is not acceptable to present the results after the introduction, without mentioning the approach you have used to obtain them.

This is due to the Nature Communications manuscript format.

2. Please provide details of the experimental equipment used for the presented experiments, as well as the design of the experiments. You can use a table to organize these.

We have included this information in the manuscript, in particular the Methods section and in the Supplementary Information. The VAM printers are home built and are described briefly in the Methods section as well as in references therein.

Diffusion in VAM

1. The name of this section is not proper. Please improve the structure as per comment 1. Additionally, consider numbering the sections.

We respectfully disagree. We feel that this is an appropriate name for this Results Subsection.

2. Some parts of this section are largely descriptive and they would be better moved in the introduction. Please point out the challenges you aim to overcome in the introduction.

We respectfully decline to do so and have kept the content organization here unchanged. Note we are following the typical structure of a Nature Communications paper which tends to have introductory and descriptive information in the Results section.

3. “This threshold exposure time is... respectively”: Please use a reference if this is taken from a source or describe the experimental setup and procedure used for this measurement.

This is simply the amount of time it takes to print the disk. We believe lines 113-119 appropriately describe this measurement.

4. Provide a description of the methods you use in a cohesive manner adding an approach section. The use of schematics is also encouraged, to facilitate the readers’ understanding and improve the visibility of your study.

This information is included at length in the main manuscript, Methods and Supplementary information. Note we are confined to the Nature Communications format.

5. Equation 1 and wherever it appears: Within equations, numbers, punctuation, parentheses, common function names, units, or mathematical signs are set upright; variables are set in italics, and vectors are set in bold. Please refer to ISO80000-1. Please make the necessary corrections in all equations.

We appreciate this comment, however this is an editorial concern and we will await editorial guidance on equation formatting.

6. Fig.2: The caption of figure 12 is over ten lines. Please describe the figures in the main text of the manuscript and keep the captions short.

This length is common for Nature Communications and similar journals.

Optical point spread function

1. As per comment 8. This applies to all section titles of this study.

Unclear which comment the Reviewer is referring to. We believe this is an appropriate subsection title.

2. Eq.(4): Consider using stacked fractions for large equations.

We will await editorial guidance on equation formatting.

3. Please correct any language-related mistakes like this (line 216): “These curves are not fits to the data”.

This language is correct. These curves are indeed not fits to the data, but rather simulation curves using physical parameters measured in separate experiments (Diffusion in VAM and Optical point spread function).

4. Line 218: “Fig.S3”. There is no such figure. Please correct this for the rest mentions of the figures as well.

Figure S3 is found in the Supplementary Information.

Correction

1. As per comment 8. This applies to all section titles of this study.

We politely decline to change the name of this subsection. This is an appropriate subsection title.

2. Line 244-248: Please elaborate further on how you concluded that smaller features require more exposure time or provide a reference.

Please see Figs. 1,3,4.

3. Lines: 273-275: Please consider adding a flowchart-schematic of the described steps.

We politely decline to insert a flowchart. However, the included Python example code will give the interested reader a working implementation to experiment on.

Correction results

1. Provide details of the experimental setup as well as the design of the experiments used.

These are provided in the Methods section.

2. The term “Benchy print” is not formal. Consider using “benchmark test artifacts” or a similar term.

This print has been removed. However, we note that “Benchy” is the official name of this benchmark artifact.

3. Line 327: “designed with nTopology, USA” please provide a reference.

This is the typical way to reference commercial software used as a tool.

4. Fig.5: The caption of figure 5 is over 5 lines. Please describe the figures in the main text of the manuscript and keep the captions short.

This is appropriate and common in Nature Communications.

5. The correction you propose seems interesting. Please provide more detailed measurements and comparisons between the four different kinds of parts: microgravity-, microgravity corrected-, earth-, and earth corrected-parts. Additionally, a comparison of the manufactured parts with the part design should take place, to point out the maturity of this AM process.

In this revision, we have added significant detail regarding quantitative comparison between manufactured and printed parts. Please see response to above reviewers for details.

Conclusions

1. Many pages in this section are not conclusions; a literature review is also included in the conclusions section. Please move those in the introduction. Please refer to comment 1.

The literature review has been moved to the Introduction. This section is titled "Discussion and Conclusion", hence the non-concluding remarks.

2. Do not summarize the entire paper in this section, but rather the results and the discussion. Also, give some hints for future work.

We believe the content in the Discussion and Conclusions section is appropriate.

3. After the conclusions, the acknowledgments, references, appendices, etc. should follow, and then the paper should come to an end. However, in this case, more sections follow, namely "Methods". Please correct this. Please refer to comment 1.

This is due to the Nature Communications format.

Methods

1. The section "Approach" or "Method" describes the theoretical framework that the paper suggests and does not necessarily refer to a specific case. An implementation and/or experimental setup should include such specified details.

We believe that the methods section is appropriate for Nature Communications format.

2. You mention that simulations took place. Please elaborate on the simulation approach and software used.

This is included in the Polymerization time simulations subsection of the Methods section. We have added a note that the simulations were performed in Python.

3. Consider organizing the experimental data you provide in the "Printing" subsection in a table.

Such a table has been added to the Supplementary Information (Tables S1, S2). Also, an Experiment Design section has been added to the Supplementary Information.

References

The manuscript could benefit from the following studies in terms of completeness of the conducted research:

Shusteff (2017) is already included as reference 7. We appreciate the further list of references, however, we do not see them as relevant for this manuscript.

- Foteinopoulos, P., Papacharalampopoulos, A., & Stavropoulos, P. (2018). On thermal modeling of Additive Manufacturing processes. *CIRP Journal of Manufacturing Science and Technology*, 20, 66-83.
- Salvekar, A. V., Nasir, F. H. B. A., Chen, Y. H., Maiti, S., Ranjan, V. D., Chen, H. M., ... & Huang, W. M. (2022). Rapid Volumetric Additive Manufacturing in Solid State: A Demonstration to Produce Water-Content-Dependent Cooling/Heating/Water-Responsive Shape Memory Hydrogels. *3D Printing and Additive Manufacturing*.
- Stavropoulos, P., & Foteinopoulos, P. (2018). Modelling of additive manufacturing processes: a review and classification. *Manufacturing Review*, 5, 2.
- Shusteff, M., Browar, A. E., Kelly, B. E., Henriksson, J., Weisgraber, T. H., Panas, R. M., ... & Spadaccini, C. M. (2017). One-step volumetric additive manufacturing of complex polymer structures. *Science advances*, 3(12), eaao5496.
- Stavropoulos, P., Foteinopoulos, P., & Papapacharalampopoulos, A. (2021). On the impact of additive manufacturing processes complexity on modelling. *Applied Sciences*, 11(16), 7743.
- Salvekar, A. V., Nasir, F. H. B. A., Chen, Y. H., Maiti, S., Ranjan, V. D., Chen, H. M., ... & Huang, W. M. (2022). Rapid Volumetric Additive Manufacturing in Solid State: A Demonstration to Produce Water-Content-Dependent Cooling/Heating/Water-Responsive Shape Memory Hydrogels. *3D Printing and Additive Manufacturing*.
- De Beer, M. P., Van Der Laan, H. L., Cole, M. A., Whelan, R. J., Burns, M. A., & Scott, T. F. (2019). Rapid, continuous additive manufacturing by volumetric polymerization inhibition patterning. *Science advances*, 5(1), eaau8723.
- Obaton, A. F., Lê, M. Q., Prezza, V., Marlot, D., Delvart, P., Huskic, A., ... & Cayron, C. (2018). Investigation of new volumetric non-destructive techniques to characterise additive manufacturing parts. *Welding in the World*, 62(5), 1049-1057.

REVIEWERS' COMMENTS

Reviewer #1 (Remarks to the Author):

The authors provided convincing new experimental measurements showing that the deconvolution provides an improved print especially when very different negative feature size form an 3d object. All questions were answered satisfactorily.

Reviewer #2 (Remarks to the Author):

The manuscript has significantly improved. Reviewers comments are properly answered and utilized to improve the manuscript. In my opinion this manuscript is in acceptable level.

Reviewer #3 (Remarks to the Author):

I appreciate the authors' efforts in revising the original manuscript. I find the inclusion of the additional case study geometries and quantitative comparisons to be especially well-thought out. I have no further comments to add at this time.